# Experimental Study on Basalt Fiber Crack Resistance of Asphalt Concrete Based on Acoustic Emission

**DOI:** 10.3390/ma14154096

**Published:** 2021-07-23

**Authors:** Kang Yang, Zhaoyi He, Dongxue Li, Hao Xu, Lin Kong

**Affiliations:** 1School of Civil Engineering, Chongqing Jiaotong University, Chongqing 400074, China; 622180111032@mails.cqjtu.cn (H.X.); konglin@mails.cqjtu.edu.cn (L.K.); 2College of Traffic & Transportation, Chongqing Jiaotong University, Chongqing 400074, China; hzyzwb@cqjtu.edu.cn (Z.H.); lidongxue@cqjtu.edu.cn (D.L.)

**Keywords:** asphalt concrete, acoustic emission, basalt fiber, crack resistance

## Abstract

In this study, the semicircle three-point bending tests of ordinary asphalt concrete and basalt fiber asphalt concrete were carried out and acoustic emission parameters were collected during the test. The differences of the characteristics of acoustic emission parameters between basalt fiber asphalt concrete and ordinary asphalt concrete were analyzed, and the damage stages were divided based on the variation of acoustic emission parameters; Rise Angle and Average Frequency were introduced to study the cracking mode and crack resistance mechanism of asphalt concrete with basalt fiber. The results show that the acoustic emission parameters can well represent the toughening and crack resistance effect of basalt fiber in asphalt concrete, and the damage stages can be divided into three stages: microcrack initiation stage, fracture stage, and residual stage. The duration of the fracture stage and the load resistance time of the specimen were greatly prolonged. The proportion of shear events in the whole failure process increased greatly after the basalt fibers were added, especially in the fracture stage, which reduced the tensile failure tendency of the specimens, and thus improved the bending and tensile performance of the specimens and played a toughening and crack resistance role in the fracture stage.

## 1. Introduction

Under the influence of vehicle load and environment, many kinds of fracture damage such as fatigue crack, temperature crack and reflection crack, will occur in the internal structure of asphalt pavement [1]. Further development of these damages will lead to the deterioration of the structure and function of asphalt pavement and will directly affect the service life of asphalt pavement. For engineers and researchers, it is of great significance and difficulty to accurately predict the pavement crack [2]. Therefore, research on the crack resistance performance and mechanism of asphalt concrete has never stopped.

Asphalt concrete belongs to composite heterogeneous material and has a complex fracture behavior. In recent years, a lot of researches showed that many kinds of fibers including polyester, lignin, glass, and basalt fibers etc. can efficiently modify asphalt concrete. Among them, basalt fiber has been widely used in asphalt pavement as a result of its great interfacial infiltration. Compared with other types of fibers, basalt fibers can effectively improve the performances including fatigue resistance, high-temperature resistance, low temperature property, moisture stability, and mechanical properties of asphalt concrete [3,4,5,6,7]. However, the present analysis and research methods on the influence of basalt fiber on the tensile resistance of asphalt concrete are still relatively simple and monotonous; most researches are still performing single mechanical analysis [8,9]. Therefore, it is necessary to find a more reliable and intuitive research method to analyze and evaluate the cracking resistance ability of basalt fiber.

Acoustic emission (AE) technology has been proved as an effective nondestructive testing method, which is of great help in detecting tiny deformation and damage inside materials and has been gradually applied to the researches including rock, structure, and concrete in recent years [10,11,12,13,14,15,16]. The principle of AE is the transient elastic wave generated by the friction, cracking, deformation, and other activities inside the material; the elastic wave propagates to the sensors; and the signals are converted and then showed as AE parameters. 

AE technology can be used to reflect the subtle changes and evaluate the mechanical and physical behaviors inside materials at the microscopic level by analyzing the characteristics of signals. Some basic AE parameters are shown in Figure 1, including amplitude, energy, rise time, and duration.

However, there are still few studies on the application of AE to asphalt concrete, most researches still focus on the AE source location and the AE parameters characterization of the fracture behavior. Qiu studied the propagation characteristics of AE standard signal in asphalt concrete, and the accuracy of AE positioning was further improved [17]; Behzad studied the self-healing property of the asphalt concrete by AE, and the results show that the cooling cycles had a great influence on the self-healing property. AE can be used to analyze the influence of the resting time between cooling cycles on materials [18]. Qiu studied the variation characteristics of AE parameters during the beam bending test of asphalt concrete; in addition, the scale invariance of AE and the abrupt change of AE under damage were studied. The results showed that the abrupt rise of waveform fractal dimension from the minimum value was related to the critical fracture condition of asphalt concrete, and the recommended sampling frequency of AE test for asphalt concrete is 1 MHz [19,20,21]. Cai used Weibull distribution to characterize the damage evolution of asphalt concrete, in which the cumulative energy was selected as the damage variable. The results showed that Weibull parameters M and N have an obvious positive correlation with matrix porosity and interlocking index, respectively [22]. Jiao carried out compression and splitting tests on permeable asphalt concrete. The results showed that the characteristics of the AE parameter can well reflect the micro damage, and AE technology has a good prospect in the application of asphalt concrete [23,24,25]. Jacob carried out fracture tests on asphalt concrete with different recycled asphalt shingle (RAS) contents at low temperature. Compared with the control group without RAS, the mixture containing RAS had lower fracture energy, higher peak load, and higher brittle breaking temperature [26]. Yang used AE to study the fracture behavior of asphalt concrete at low temperature and location of damage points. The results showed that AE can well characterize the fracture process of low-temperature asphalt concrete, and the occurrence of location points can visually display the crack expansion path [27].

AE technology has been gradually applied to the research of ordinary asphalt concrete (OAC), however, there are few studies on fracture behavior of asphalt concrete with basalt fiber addition. It is of great value to study the fracture behavior of basalt fiber asphalt concrete (BFAC) at the micro level based on AE, so this paper aims to: Study the differences of variation characteristics and laws of AE parameters during the fracture and damage process of OAC and BFAC.Study the effect and mechanism of crack resistance of basalt fiber in asphalt concrete damage based on AE parameters.Explore the cracking mode and crack resistance mechanism of OAC and BFAC by AE parameters Rise Angle (RA) and Average Frequency (AF).

## 2. Raw Materials 

### 2.1. Asphalt and Aggregate

In this experiment, the asphalt with a 70 penetration grade was used, which was from Dongqi Petrochemical Industry of Chongqing, Chongqing, China; limestone was used as the aggregate and mineral filler. According to Standard Test Methods of Bitumen and Bituminous Mixtures of Highway Engineering (JTG E20-2011), the target aggregate gradation was designed as the mid-value between the upper and lower limits, as shown in Figure 2.

### 2.2. Basalt Fiber

As shown in Figure 3, basalt fiber with a length of 12 mm was selected in this paper and the basic properties are listed in Table 1.

## 3. Test setup and Methods

### 3.1. Preparation of Specimens

AC-13 BFAC was utilized with a grade 70 penetration bitumen content of 4.3% and mineral filler content of 6%; the fiber content was 0.5% by the weight of mixture. Referring to the relevant literature [28,29], the following preparation process was formulated: First, aggregate and asphalt were mixed for about 90 s at 160 °C to make them uniformly mixed, then basalt fiber was gradually added and also mixed for about 90 s; finally, mineral filler was added. At the same time, except for the fact that basalt fiber was not added, the material and preparation process of AC-13 OAC were the same with BFAC.

After mixing, the AC-13 was compacted at 140 °C and the cylindrical specimen with a radius of 75 mm was formed and then cut into the specimens with a thickness of 50 mm. Furthermore, a pre-cut notch was made at the bottom of the semi-circular specimens, which was of 10 mm in length and 2 mm in width; the test samples with basalt fibers added were numbered T1~T6. 

At the same time, the control samples without basalt fibers were prepared and numbered S1~S6. The formed specimens are shown in Figure 4.

### 3.2. Test Setup and AE System

In this test, a UTM-25 was utilized in the semi-circular bending test, which was manufactured by SANS in Guangdong, China. The loading rate was set at 1 mm/min until the end of the test.

As shown in Figure 5, using a 16-channel SEAU3H AE system manufactured by Soundwel in Beijing China, AE were also carried out during the test. As shown in Figure 6, four AE sensors were put on the surface of the specimen, which has a receiving frequency ranging from 18 to 180 kHz. The preamplifier was 40 dB and the threshold value was 35 dB, the sampling frequency was 1 MHz for the tests.

### 3.3. Rise Angle (RA) and Average Frequency (AF)

As a significant research method of AE, rise angle (RA) and average frequency (AF) have been proved to be the key parameters to characterize the tensile and shear modes [20,29], and they are obtained from the equation.
RA = Rise Time/Amplitude, ms/V(1)
AF = Counts/Duration, kHz(2)

As shown in Figure 7 and Figure 8, when the specimen is damaged, there will be different fracture modes including tensile crack and shear crack occurring inside the material; what’s more, the different shapes of AE waveforms are associated with different fracture modes. Tensile modes are corresponding to high energy, high amplitude, and have a short rise time and duration time, leading to lower RA and higher AF value. Conversely, shear modes show low frequency and longer rise time and duration time, resulting in higher RA and lower AF value [30].

## 4. Crack Resistance Analysis

### 4.1. Analysis Based on Fracture Energy

As can be seen from Figure 9, the basalt fibers on the fracture section were marked by red circles. A large number of basalt fibers were well-distributed and exposed on the fracture section, indicating that the basalt fibers had great internal dispersion and were uniformly distributed in the specimen before fracture, which is of great help in improving the crack resistance performance of the specimen. 

From the perspective of fracture mechanics, the fracture energy is used to represent the work done by unit area crack expansion to unit length. In the semi-circular bending test, the area enclosed by the load-deflection curve and the coordinate axis is the fracture energy [31]. The failure load and fracture energy of S1–S6 of OAC and T1–T6 of BFAC are shown in Figure 10. 

As can be seen from Figure 10, the average failure load of BFAC group was 4.86 kN, while that of OAC group was 4.76 kN. There were no obvious differences between the two groups on the failure load, and the addition of basalt fibers did not show the great improvement effect on the bending failure load as expected. In terms of fracture energy, the average fracture energy of the BFAC group was 11.24 kN·mm, while that of the OAC group was 9.02 kN·mm. The average fracture energy of the BFAC group was significantly higher than that of the OAC group and the average fracture energy increased by 24.6%, indicating that the load curve of BFAC dropped more slowly and covered a larger area compared with OAC; the asphalt concrete with basalt fibers added has better crack resistance effect after failure load, which is the propagation process of macrocrack.

### 4.2. Analysis Based on AE Parameters

For AE parameters, the specific values varied with different specimens while the variation trends and rules were similar and consistent. In this paper, S1 of OAC and T2 of BFAC specimens are taken as representative examples for discussion and analysis.

As shown in Figure 11, basic AE parameters of OAC and BFAC during the test are collected and analyzed, including duration, ringing counts, and cumulative ringing counts, which can reflect the number of AE activities, while amplitude, energy and cumulative energy reflect the strength [32,33,34,35]. Synthetically analyzing and considering the variation characteristics of each AE parameter, the whole process of OAC and BFAC fracture damage can be divided into three stages: I: microcrack initiation stage, II: rapid fracture stage, III: residual stage.

Microcrack initiation stage: During this stage, the time domain of OAC was 0~123 s and 0~115 s of BFAC and the load curve showed a linear growth with the increasing displacement. The variation characteristic of energy and ringing counts of OAC and BFAC were highly similar in this stage, which were maintained at a low level and the AE signals had a short duration and low amplitude. There was no obvious upward trend for the cumulative ringing counts and the cumulative energy curve, indicating that only sporadic AE activities were produced and low-strength signals were collected during this stage. These AE signals were generated by the original internal voids being continuously squeezed and compacted when the specimen was subjected to external load. The stress was released at the tip of the original crack inside, leading to the occurrence of micro-crack. At this stage, the variation characteristics and trend of AE parameters of OAC and BFAC have a high similarity, indicating that basalt fiber did not play a role in this stage.

Rapid fracture stage: In this stage, the load reached peak value and the corresponding time domain of OAC and BFAC was 123~173 s and 115~212 s, respectively; the duration time of BFAC was 97 s, which was obvious longer than 50 s of OAC. At the initiation of this stage, the energy and ringing counts of OAC and BFAC showed mutations of different degrees. The cumulative energy and cumulative ringing count curve of OAC had a sudden rise and then rose steadily. The rising rate of the cumulative energy and cumulative ringing count curves of BFAC had increased significantly compared with the previous stage, however, it is still slower than that of OAC in this stage, and there was no steep increase point similar to that of OAC. What is more, the specific values of the cumulative energy and the cumulative ringing count of BFAC were much higher than that of OAC in this stage.

The distribution of high energy and high ringing count of BFAC was also more intensive than that of OAC, and the number of AE signals with high amplitude and long duration increased significantly, indicating that the number of high-energy AE events of BFAC is significantly more than that of OAC. The BFAC AE events showed better continuity in terms of time domain, and the numbers of AE activities were also more evenly distributed compared with OAC.

Analyzing the reason, at this stage, the stress level in the middle and lower tensile zone of the specimen increased significantly with the increase of vertical displacement. The first crack point occurred at the precut crack of the specimen. The stress accumulated in the OAC, and BFAC during the previous stage was released at the internal weak points, leading to the occurrence of macrocracks that gradually propagated to the compression zone. 

During the process of the crack propagation upwards, the basalt fibers inside the material played a bridging effect role at the crack tip, and the friction was produced due to the relative sliding between the fibers and the asphalt, resulting in the improvement of the ability to resist post-peak load of specimens. Macroscopically, the decreasing rate of the load curve was delayed and the propagation rate of crack was obviously slowed down. At the same time, the fracture of basalt fibers and the friction between the fibers and the asphalt led to the generation of more and stronger AE activities and denser and evener AE signals. The basalt fibers also played the role of crack resistance in this stage.

Residual stage: The increasing vertical displacement in this stage led to the crack propagating upwards and the specimen was severely damaged and lost bearing capacity. The energy level in this stage was already very low and the numbers were few, resulting in the low level AE parameters of OAC and BFAC. The cumulative energy and cumulative ringing count curves tended to be flat, and the weak friction between aggregates produced sporadic low-energy AE events.

As can be seen from Figure 11, the addition of basalt fibers could inhibit the propagation of the macroscopic main crack and exert the effect of toughening and cracking resistance. Moreover, with the addition of basalt fibers, the AE activities inside the specimen were more abundant and AE signals were stronger. However, the basalt fibers played a role in the stage lag behind the specimen itself bearing the load, so they had no effect on improving the bending failure load.

## 5. Fracture Mode Analysis

Furthermore, in order to analyze and study the inhibition effect and mechanism of fibers on crack diffusion of asphalt concrete from the perspective of AE, rise angle (RA) and average frequency (AF) were introduced to analyze the fracture mode of OAC and BFAC. RA and AF values in the whole process of OAC and BFAC fracture damage were calculated and the distribution was obtained. 

To better study the influence of basalt fibers on the fracture mode, the main effective stage of fibers, namely the rapid fracture stage, was further divided into several periods by equal time. The rapid fracture stage of OAC was further evenly divided into seven stages, while BFAC was eight stages. According to [30], the slope of boundary line was selected as 0.5 based on the minimum dissimilarity of clustering principle, thus, the boundary line was set as AF/RA = 0.5. The RA and AF distribution at different stages of OAC and BFAC were shown in Figure 12 and Figure 13.

The RA and AF distribution of OAC are shown in Figure 12. At the microcrack initiation stage, only a small amount of AE signals with low RA and AF occurred, indicating that the damage degree inside the specimen was low at this stage, and the AE signals were mainly generated by the compaction of internal voids. During the rapid fracture stage, due to the occurrence and propagation of the main crack, the AE signals increased significantly, most of which gathered in the tensile region of RA within the range of 0~300 ms/V and AF within 0~250 kHz, while only a few signal points were in the shear region, indicating that the damage mode of specimen at this stage was mainly tensile fracture.

Until the last residual stage, there were still a large number of AE signal points that fell into the tensile region; among the whole points, 87% of the damage modes belong to tensile events, while only 13% belong to shear events. Therefore, in the whole fracture process of OAC, tensile modes play a dominant role.

The RA and AF distribution of BFAC is shown in Figure 13. It can be seen from the figure that the number of AE signal points of BFAC was much more than that of OAC during the whole fracture process. The distribution characteristic of BFAC was consistent with OAC during the microcrack initial stage, only a small amount of AE signals with low RA and AF occurred. After entering the rapid fracture stage, a large number of shear events occurred. Compared with OAC, although there were still a large number of signal points falling into the tensile region of RA within 0–400 ms/V and AF within 0–600 kHz, the number of signal points in the shear region of RA with 0–800 ms/V and AF within 0–250 kHz in this stage significantly increased, and the number of AE signal points with high RA also increased. What is more, AE signal points were more dispersed during this stage, unlike OAC signal points that were too concentrated.

After the specimen was completely damaged, the tensile events still dominated 72% while the amount of shear events accounted for 28%, much more than 13% of OAC. 

Analyzing the reason, with the addition of basalt fibers, the internal force of the specimen in the fracture process was more complex and unpredictable; the existence of basalt fibers made fracture modes more abundant, which was no longer just the tensile fracture of absolute dominance. The significant increased proportion of the shear events reduced the proportion of tensile events, which play a leading role in the fracture process and are exactly what we want to avoid in pavement engineering. Combined with these factors, the bending and tensile failure tendency of asphalt concrete was reduced with basalt fibers added.

From the perspective of RA and AF, it can be found that the effect of basalt fibers on the toughness and crack resistance mainly occurred in the rapid fracture stage, which was consistent with the analysis results based on the AE parameter. Moreover, the addition of basalt fiber can reduce the proportion of tensile fracture modes, which was also one of the reasons for the toughness and crack resistance effect. In addition, compared with AE basic parameters, RA and AF values can more intuitively and deeply reflect the fracture modes inside material. Therefore, more attention should be paid to the analysis of RA and AF values in the following analysis and research.

In this paper, the crack resistance effect of AC13 BFAC was studied using the same research methods and conclusions that apply to other types of asphalt concrete. However, it is worth noting that the differences in materials and the test method will lead to the corresponding differences in specific value and data of experimental results. Therefore, in the analysis of other type of materials and the type of test, more tests validation and data analysis should be carried out, so as to have a complete validation of the process.

## 6. Conclusions

In this study, based on OAC and BFAC asphalt concrete semicircle bending tests and AE tests, the AE parameters and the fracture modes during the cracking process of specimens were analyzed. The main conclusions are as follows:(1)The AE technology can effectively reflect the influence of basalt fibers on the damage of asphalt concrete material from the microscopic level, and basalt fibers played a role in the stage lag behind the specimen itself, bearing the load. So they had no improvement effect on the bending failure load.(2)The whole process of OAC and BFAC fracture damage can be divided into three stages based on AE parameters: microcrack initiation stage, rapid fracture stage, and residual stage. Among the three stages, the second stage had the greatest difference between OAC and BFAC. In this stage, the AE parameter of BFAC had larger values, more quantities, more dense occurrence, and evener distribution, which is also the main stage when basalt fibers played the role of crack resistance and toughening.(3)The fracture modes analysis shows that RA and AF values can more intuitively and deeply reflect the fracture modes inside material. The existence of basalt fibers made the shear events increase significantly and fracture modes more abundant; furthermore, the tensile fracture tendency was reduced and the effect of basalt fibers on crack resistance and toughness was reflected from the side.

## Figures and Tables

**Figure 1 materials-14-04096-f001:**
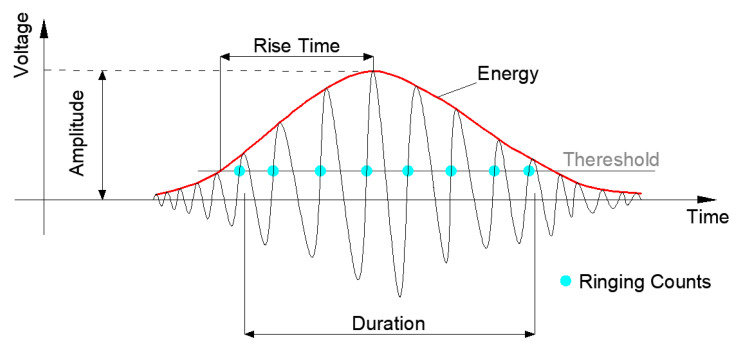
Typical AE signal.

**Figure 2 materials-14-04096-f002:**
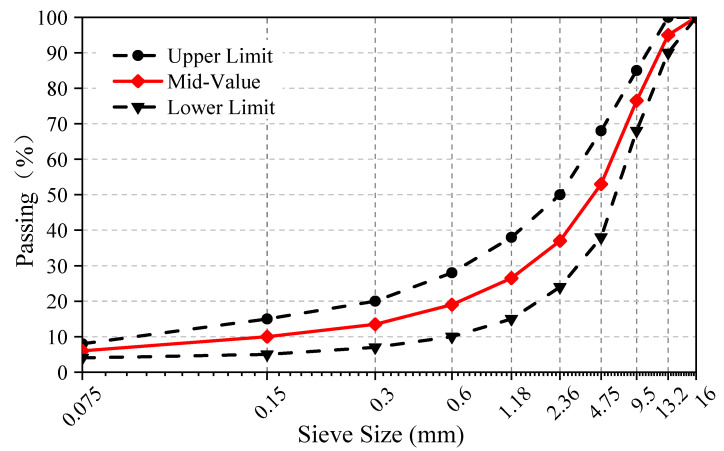
Gradation curve for AC-13.

**Figure 3 materials-14-04096-f003:**
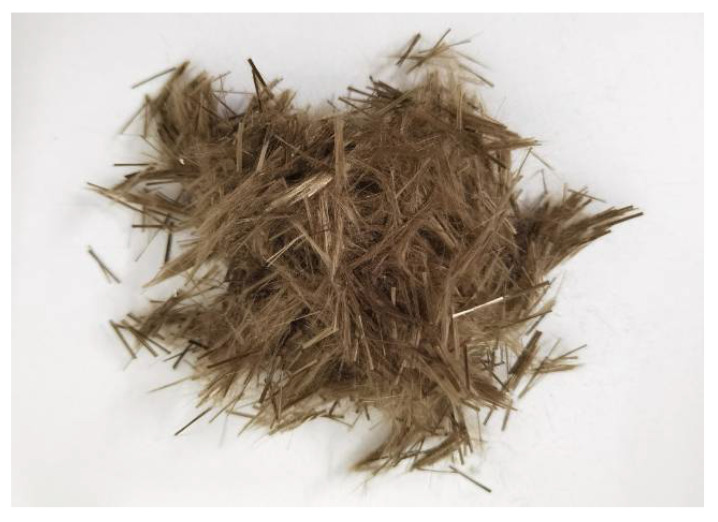
Basalt fiber.

**Figure 4 materials-14-04096-f004:**
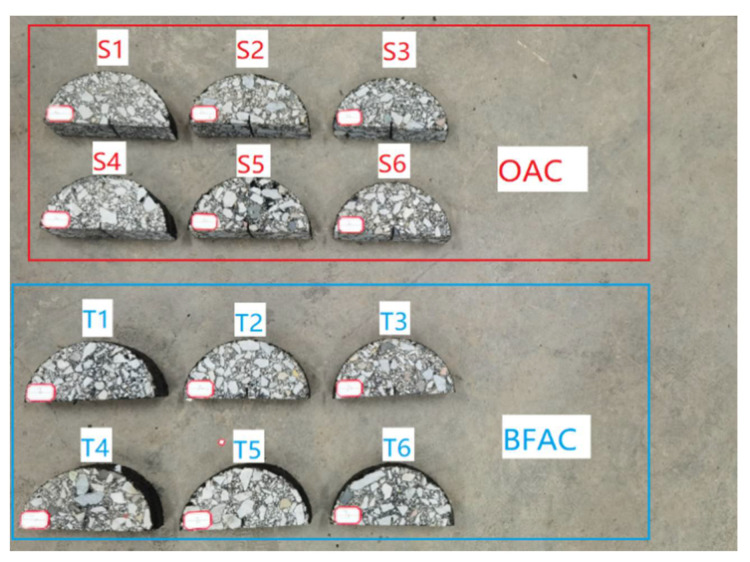
Formed specimens.

**Figure 5 materials-14-04096-f005:**
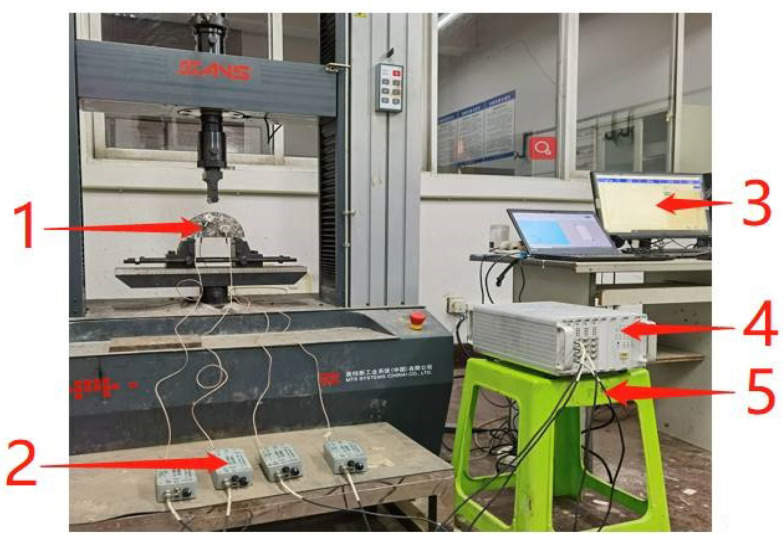
AE test system: 1: G8 sensors; 2: Preamplifier; 3: Computer; 4: High-speed digital AE detector; 5: Cable.

**Figure 6 materials-14-04096-f006:**
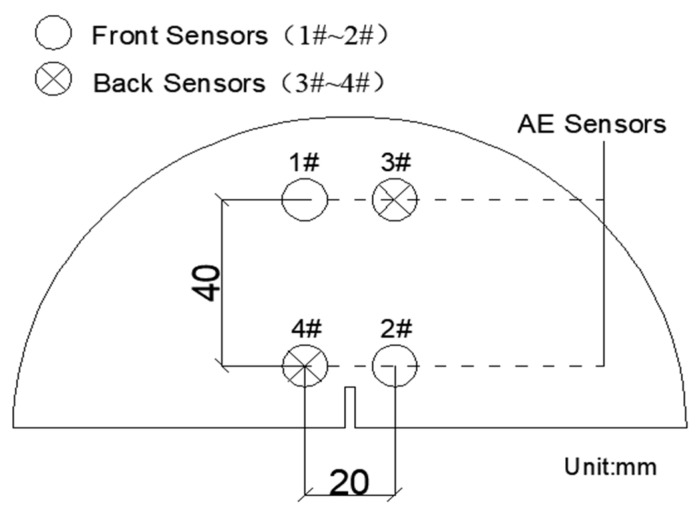
Layout of sensors.

**Figure 7 materials-14-04096-f007:**
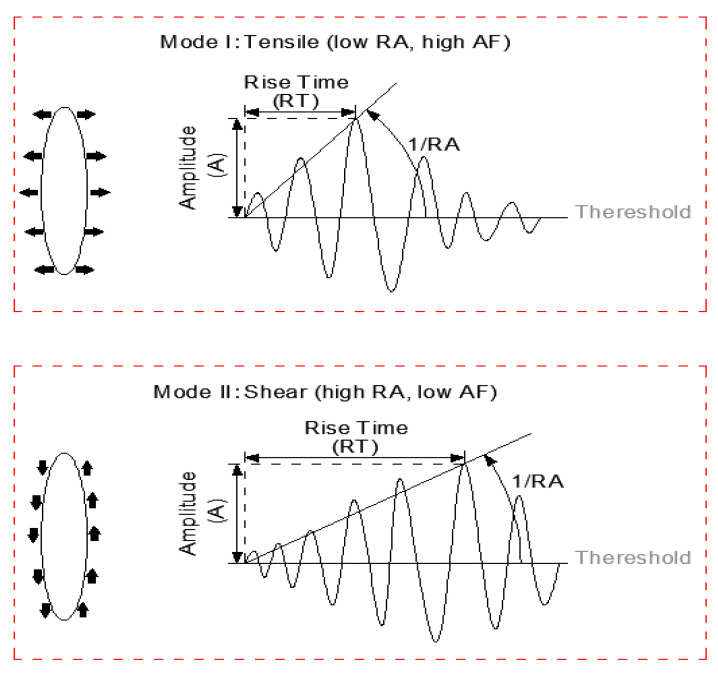
Typical fracture modes and corresponding AE waveforms.

**Figure 8 materials-14-04096-f008:**
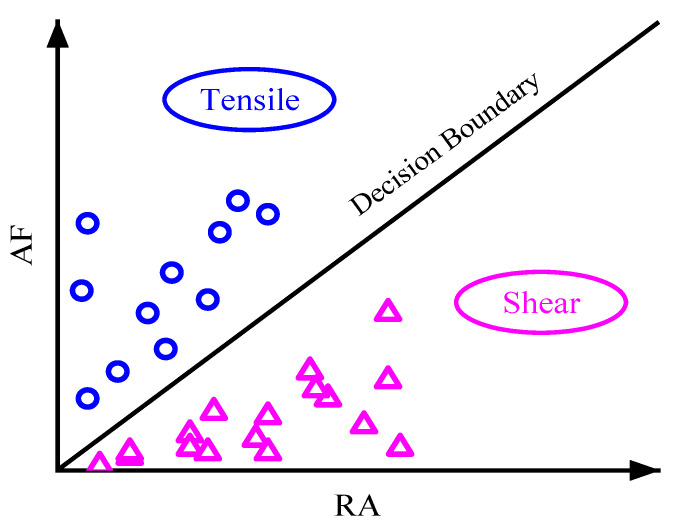
Typical fracture modes based on RA and AF.

**Figure 9 materials-14-04096-f009:**
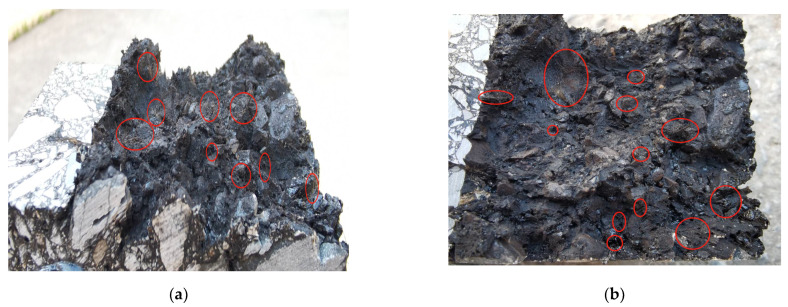
The basalt fibers on the fracture section: (**a**) Specimen T1, (**b**) Specimen T2.

**Figure 10 materials-14-04096-f010:**
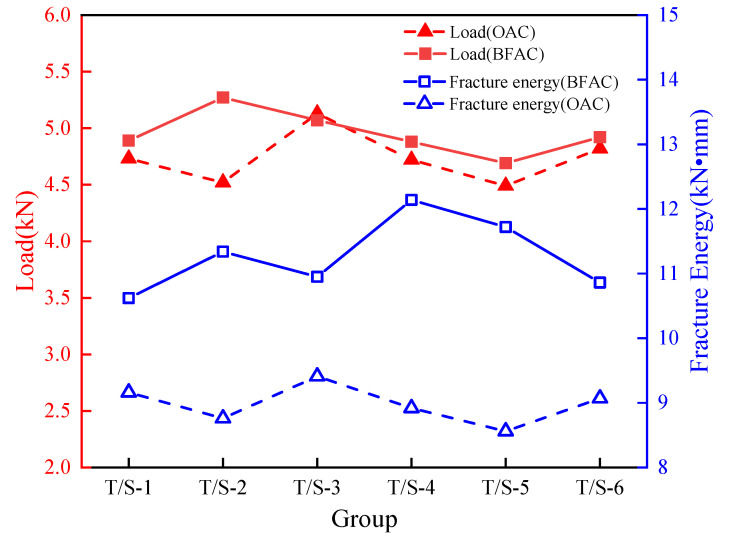
The maximum load and fracture energy of different specimens.

**Figure 11 materials-14-04096-f011:**
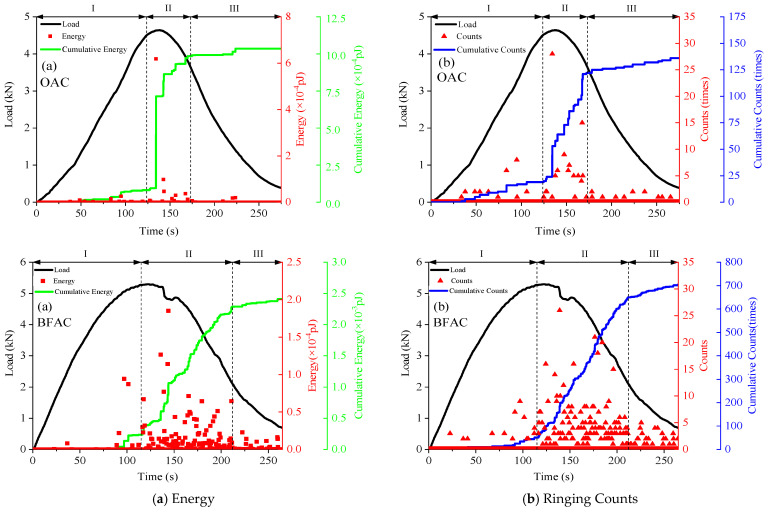
Characteristics of AE parameters of OAC and BFAC; (**a**) Energy; (**b**) Ringing Counts; (**c**) Amplitude; (**d**) Duration.

**Figure 12 materials-14-04096-f012:**
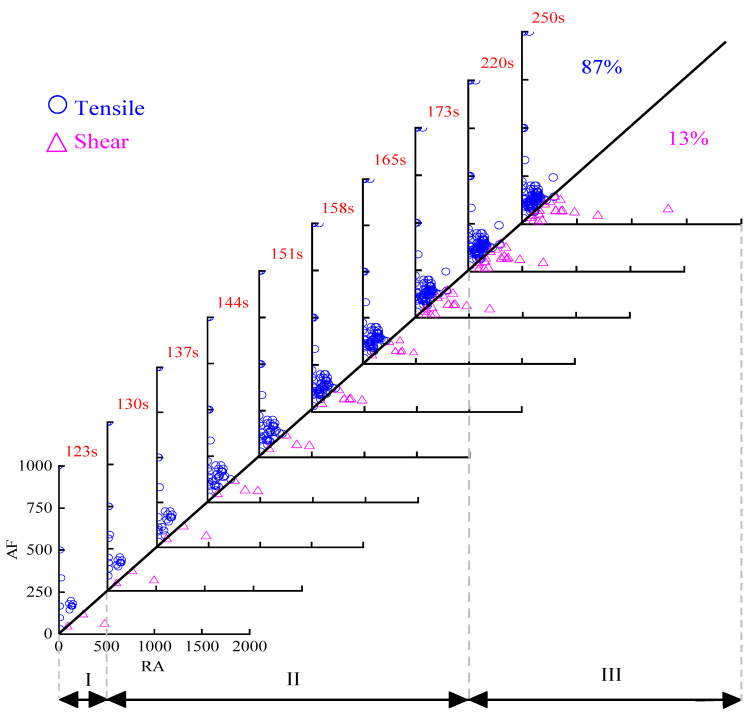
RA and AF distribution of OAC.

**Figure 13 materials-14-04096-f013:**
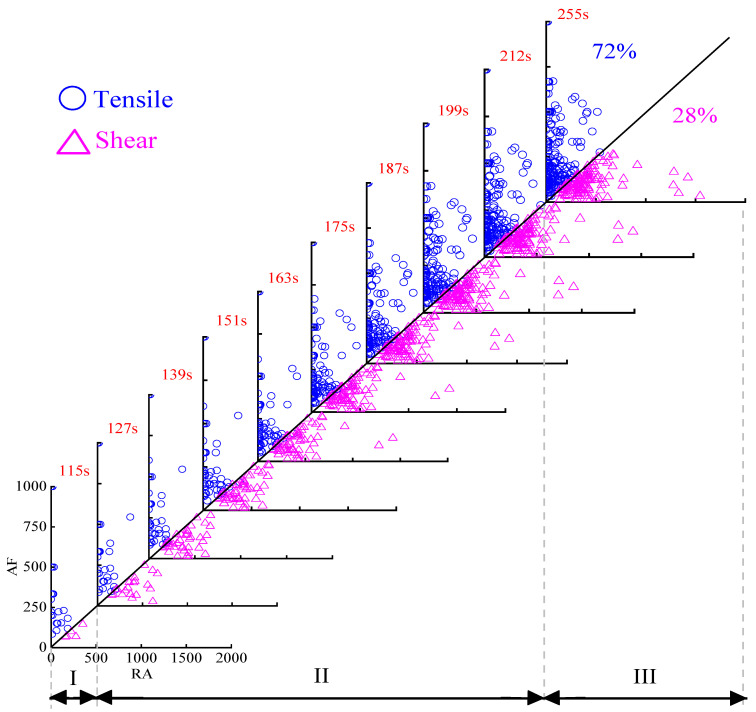
RA and AF distribution of BFAC.

**Table 1 materials-14-04096-t001:** Basic properties of basalt fiber.

Index	Diameter (um)	Specific Gravity (g/cm^3^)	Tensile Strength (Mpa)	Elasticity Modulus (Gpa)	Elongation (%)
Value	13	2.7–3.1	2700	65	2.5–2.8

## Data Availability

The data used to support the findings of this study are included within the article.

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
