# Peer review of "Experimental Study on Basalt Fiber Crack Resistance of Asphalt Concrete Based on Acoustic Emission"

_materials, 2021, doi:10.3390/ma14154096_

Round 1

Reviewer 1 Report

Please address my comments:

Abstract: The abstract should be a total of about 200 words maximum, according to Instructions for authors. Yours has 246 words.

General comment:

The authors prepared a comprehensive research framework, well-described and with a sharp interpretation of the results.
However, some of the inferences could be influenced by the type of material (limestone AC-13) and the type of test. So it is advisable in the Conclusion (section 5) to underline this aspect, pointed out that the underlined analysis methodology (AE) should experiment for other types of AC (other aggregates and another grading) and other types of fatigue tests to have a complete validation of the process.

Author Response

Thank you very much for your positive comments. Please refer to our replies in the attachment.

Reviewer 2 Report

The originality and the scientific value of the subject research are good.
The research area is experimental study on basalt fiber crack resistance of asphalt concrete based on acoustic emission.

The research area is very interesting and also current.
The research includes an experimental program.

The manuscript has the usual structure, but part of the discussion must be indicated separately.
The Number of Pages is incorrect. (In abstract in MDPI)
The overall of the informative value must be improved. 

Extensive research is underway in the area of concrete structures/materials and measurement of acoustic emission when it is necessary to rework and expand the information in the introduction section. 
These are mainly examples and information of measurement of acoustic emission:
Pazdera, L., et. al. Measurement and utilization of acoustic emission for the analysis and monitoring of concrete slabs on the subsoil. Periodica Polytechnica Civil Engineering 2019, 63 (2), 608-620.

and many others.

Create a list of shortcuts. The document will be clearer.
Enlarge Figure 7.
Figure 10 is on two pages.
Improve the informative value of Figures 10 and 11.
Were further tests of mechanical properties performed? compressive strength? modulus of elasticity? larger test specimens?
State the relations for the calculation of fracture energy.

Enlarge Figure 2.
Enlarge Figure 4.
Enlarge Figure 5.

Provide more (graphical) outputs from the measurement. 
Overall, it is necessary to process the manuscript with greater interest.
The discussion chapter must be presented separately and present the results in the context of current research. What is the same, what is different?

Overall, it is necessary to improve the presentation of the results of experiments and increase the informative value of the results.
The manuscript must be revised.

Author Response

(The authors gave the same response as above.)

Reviewer 3 Report

  • The language of the manuscript has to be improved. 
  • The abstract needs improvement. It should begin with a short introduction, followed by objectives, and concluded with major conclusions. 
  • Your introduction has a lot of useful information. I missing some general information regarding the performance of asphalt mixtures such as doi.org/10.1080/14680629.2017.1283353 and doi.org/10.1080/14680629.2021.1908408
  • The objectives are clearly stated.
  • For section 2.1, provide the basic properties of asphalt and aggregates.
  • How have you determined bitumen content of 4.3%?
  • Provide mix design data (air voids, VMA, VFA, etc)
  • How have you determined fiber content?
  • How have you determined mixing and compaction temperature, and how many gyrations have you used to compact specimens?
  • The conclusions look good.

Author Response

(The authors gave the same response as above.)

Round 2

Reviewer 2 Report

Thank you for the adjustments made.
The changes made the improvement of the manuscript.

The research area and results are from the context of the manuscript can better understand.

The manuscript contains all the main information.

Reviewer 3 Report

Thank you for addressing my comments.